# Glycated Walnut Meal Peptide–Calcium Chelates: Preparation, Characterization, and Stability

**DOI:** 10.3390/foods13071109

**Published:** 2024-04-04

**Authors:** Zilin Wang, Ye Zhao, Min Yang, Yuanli Wang, Yue Wang, Chongying Shi, Tianyi Dai, Yifan Wang, Liang Tao, Yang Tian

**Affiliations:** 1College of Food Science and Technology, Yunnan Agricultural University, Kunming 650201, China; wangzilin928@163.com (Z.W.); zhaoye202404@163.com (Y.Z.); miny1115@163.com (M.Y.); wangyuanli19971230@163.com (Y.W.); wangyue981117@163.com (Y.W.); shichongying@163.com (C.S.); dtynongda@163.com (T.D.); w2684159508@163.com (Y.W.); 2Engineering Research Center of Development and Utilization of Food and Drug Homologous Resources, Ministry of Education, Yunnan Agricultural University, Kunming 650201, China; 3Yunnan Key Laboratory of Precision Nutrition and Personalized Food Manufacturing, Yunnan Agricultural University, Kunming 650201, China; 4Puer University, Puer 665000, China

**Keywords:** walnut meal protein hydrolysates, glycated peptide–calcium chelates, physicochemical properties, structural characterizations, stability

## Abstract

Finding stable and bioavailable calcium supplements is crucial for addressing calcium deficiency. In this study, glycated peptide–calcium chelates (WMPHs–COS–Ca) were prepared from walnut meal protein hydrolysates (WMPHs) and chitosan oligosaccharides (COSs) through the Maillard reaction, and the structural properties and stability of the WMPHs–COS–Ca were characterized. The results showed that WMPHs and COSs exhibited high binding affinities, with a glycation degree of 64.82%. After glycation, Asp, Lys, and Arg decreased by 2.07%, 0.46%, and 1.06%, respectively, which indicated that these three amino acids are involved in the Maillard reaction. In addition, compared with the WMPHs, the emulsifying ability and emulsion stability of the WMPHs–COS increased by 10.16 mg^2^/g and 52.73 min, respectively, suggesting that WMPHs–COS have better processing characteristics. After chelation with calcium ions, the calcium chelation rate of peptides with molecular weights less than 1 kDa was the highest (64.88%), and the optimized preparation conditions were 5:1 *w*/*w* for WMPH–COS/CaCl_2_s, with a temperature of 50 °C, a chelation time of 50 min, and a pH of 7.0. Scanning electron microscopy showed that the “bridging role” of WMPHs-COS changed to a loose structure. UV–vis spectroscopy and Fourier transform infrared spectrometry results indicated that the amino nitrogen atoms, carboxyl oxygen atoms, and carbon oxygen atoms in WMPHs-COS chelated with calcium ions, forming WMPHs-COS-Ca. Moreover, WMPHs-COS-Ca was relatively stable at high temperatures and under acidic and alkaline environmental and digestion conditions in the gastrointestinal tract, indicating that WMPHs–COS–Ca have a greater degree of bioavailability.

## 1. Introduction

Walnuts (*Juglans regia* L.) are rich in nutrients, such as protein, carbohydrates, unsaturated fatty acids, vitamin A, and vitamin E [1]. China is a major walnut producer globally, with a significant number of walnut plantations located in Yunnan and Xinjiang [1]. Because the oil content of walnuts is more than 60% and because of the high content of unsaturated fatty acids, including linoleic and linolenic acids, most walnuts are used primarily for oil extraction, which leads to the production of walnut meal [2,3]. Walnut meal is composed of more than 40% protein with a high essential amino acid content and can be used to obtain biological peptides, making it a nutritious and inexpensive source of high-quality plant protein [3,4]. However, most walnut meal is commonly used as a fertilizer or animal feed, resulting in the waste of high-quality protein resources [5]. Therefore, the increased utilization and economic value of walnut meal have received widespread attention.

Calcium is a vital nutrient in the human body that plays a crucial role in intracellular metabolism, heart function, and bone development and is found primarily in bones and teeth [6]. Calcium deficiency can lead to various health issues, such as rickets, high blood pressure, and osteoporosis [7]. Based on current dietary habits, joint calcium supplements such as calcium carbonate and calcium lactate can form insoluble salt precipitates with phytic acid, oxalic acid, and other substances in the alkaline environment of the intestinal tract, which significantly restricts the absorption and utilization of calcium in the body [8]. Due to the aging population, there is an urgent need to develop effective calcium supplements to prevent and treat bone diseases. Bioactive peptides derived from protein hydrolysates can act as carriers to deliver calcium to various body parts and fully utilize their functions [9]. The novel calcium supplements, food-derived peptide and calcium chelates, open calcium channels in cell membranes and promote calcium absorption [10]. Peptide–calcium chelates derived from tilapia skin [11] and sheep bone collagen [12] can enhance calcium absorption in the body and improve the bioavailability of calcium, suggesting that peptide–calcium chelates can be advantageous for calcium absorption.

The Maillard reaction, also known as glycation, is a spontaneous cascade reaction that occurs between saccharides and proteins or peptides during heat treatment of food [13]. Research has demonstrated that the Maillard reaction can enhance the thermal stability, acid–base stability, and antioxidant capacity of peptides [14,15]. Recently, peptides and saccharides have been used as biopolymers in slow-release carrier assemblies to slow the reaction rate and release of various bioactive compounds in vivo [16]. COSs are easily absorbed and utilized by the small intestine due to their low viscosity, high water solubility, high biocompatibility, and biodegradability [17], making them good materials for peptide modification. Modifying peptides via the Maillard reaction is necessary to improve the stability of peptide–calcium chelates.

In the present study, walnut meal protein peptides (WMPHs) were obtained by enzymatic hydrolysis, WMPHs were combined with COSs by the Maillard reaction, and the modified products were chelated with calcium ions. The temperature, pH, and in vitro simulated digestion stability of the chelates were investigated, providing a theoretical basis for using WMPHs–COS–Ca as a new calcium supplement product with stable processing characteristics and high efficacy and safety.

## 2. Materials and Methods

### 2.1. Materials and Reagents

Walnut meal was purchased from Yunnan Morre Gaden Biotechnology Development Co., Ltd. (Chuxiong, China). A BCA protein assay kit, Coomassie blue fast staining solution (R250), and SDS–PAGE protein sample loading buffer were purchased from Shanghai Beyotime Biotechnology Co., Ltd. (Shanghai, China). Alkaline protease (enzyme activity/titer: ≥200,000 U/g) and β-mercaptoethanol were purchased from Beijing Solarbio Technology Co., Ltd. (Beijing, China). SDS–PAGE precast gels (12%) were purchased from Wansheng Haotian Biotechnology Co., Ltd. (Shanghai, China). Chitosan oligosaccharides (molecular weight: ≤1000), phthaldialdehyde (OPA), sodium dodecyl sulfate (SDS), sodium tetraborate, CaCl_2_, KCl, KH_2_PO_4_, NaHCO_3_, NaCl, Mg_2_Cl(H_2_O)_6_, and (NH_4_)_2_CO_3_ were purchased from Shanghai Aladdin Bio-Chem Technology Co., Ltd. (Shanghai, China), and NaOH, HCl, methanol, acetic acid, and glutaraldehyde were purchased from Chengdu Chron Chemical Reagent Co., Ltd. (Chengdu China).

### 2.2. Preparation of Walnut Meal Protein Hydrolysates

Walnut meal protein hydrolysate (WMPHs) were prepared according to our previous study [18]. Walnut protein was extracted by alkali extraction and acid precipitation; subsequently, the mixture was dissolved in ultrapure water after freezing–drying (2% *w*/*v*), 2% (*w*/*w*) alkaline protease was added, and 1 M NaOH was used to adjust the pH of the mixed solution to 9.0. Afterwards, the mixture was hydrolyzed at 55 °C for 3 h, heated at 100 °C for 10 min, centrifuged (4000 rpm, 20 min), frozen and dried after the liquid supernatant was collected, and stored at −40 °C for further use.

### 2.3. Preparation and Process Optimization of WMPHs–COS

#### 2.3.1. Preparation of WMPHs–COS

WMPHs and COSs were mixed in ultrapure water, thoroughly mixed for 1 h, and subsequently incubated at 4 °C for 12 h. The reaction was performed in a water bath at an appropriate temperature and quickly cooled to room temperature. Dialysis was performed in a 100–500 Da dialysis bag of ultrapure water for 24 h, followed by centrifugation (3000 rpm, 10 min). The liquid supernatant was collected and freeze-dried [19].

Degree of glycation: The protocol was conducted based on the procedure described in a previous study [20]. OPA (40.0 mg) was dissolved in 1.0 mL of methanol, 2.5 mL of 20% (*w*/*w*) SDS solution, 2.5 mL of 0.1 mol/L sodium tetraborate solution, 100 μL of β-mercaptoethanol, and 50 mL of ultrapure water. Four milliliters of OPA reagent was added to 200 μL of the glycated peptide solution in a 35 °C water bath for 2 min, after which the absorption at 340 nm was measured. The degree of glycation was calculated as follows:Degree of glycation (%) = (A_0_ − A_t_)/A_0_(1)
where A_0_ is the absorbance before the Maillard reaction and A_t_ is the absorbance after the Maillard reaction.

Degree of browning: The browning degree was determined by adding 1 mL of the glycated peptide solution to 5 mL of 0.1% SDS, mixing the mixture, and determining the absorption at 420 nm [21].

#### 2.3.2. Single-Factor Experiment

The fixed conditions were a WMPHs/COS ratio of 1:1 *w*/*w*, a time of 6 h, a concentration of WMPHs of 15 mg/mL, a temperature of 50 °C, and a pH of 8. The influence of the WMPH/COS ratio (1:3, 1:2, 1:1, 2:1, and 3:1 *w*/*w*), time (2, 4, 6, 8, and 10 h), concentration of WMPHs (5, 10, 15, 20, and 25 mg/mL), temperature (50, 60, 70, 80, and 90 °C), and pH (6.0, 7.0, 8.0, 9.0, and 10.0) on the glycation degree and browning degree of the WMPHs–COS was investigated.

#### 2.3.3. Response Surface Methodology (RSM) Optimization

According to the one-way experimental results, the Box–Behnken design (BBD) in RSM was used to optimize the process [22]. The time, concentration of WMPHs, pH, and temperature were selected as response surface factors. The glycation degree was used as the response value (*Y*), and 4-factor and 3-level experiments were conducted using the BBD design to optimize the preparation process of WMPHs–COS. The factor levels are shown in Table 1, and the experimental data listed in Table 2 were analyzed through multiple regression using Design-Expert. V8.0.6. 

### 2.4. The Structure and Processing Characteristics of WMPHs–COS

#### 2.4.1. SDS–PAGE

SDS–PAGE was conducted according to the procedure of Shi, Bi [23], with slight modifications. Briefly, the concentrations of the stacking and separating gels were 6% and 12%, respectively. The samples were heated at a constant voltage of 60 V until all the samples entered the top of the separating gel. Then, the voltage was adjusted to 120 V until the end of electrophoresis. A molecular marker in the range of 10–180 kDa was used as a standard to estimate the molecular weight of the samples. The gels were fixed with 0.5% glutaraldehyde and stained with Coomassie blue fast staining solution. The gels were viewed and photographed with a Bio-Rad Molecular Imager (XRS+, Bio-Rad, Hercules, CA, USA).

#### 2.4.2. Amino Acid Composition

Total amino acid data were detected by a Biochrom 30+ Amino Acid Automatic Analyzer (Biochrom, England). Briefly, the samples were transferred to hydrolysis tubes. Then, 10 mL of 6 M HCl was mixed thoroughly in each tube. After being filled with N_2_ for 1 min to remove air, the tubes were sealed with caps and placed at 110 °C for 24 h. The prepared solutions were filtered through a 0.45 μm filter membrane for subsequent amino acid analysis.

#### 2.4.3. Circular Dichroism (CD) Spectroscopy

A 1.0 mg sample was weighed and dissolved in 10 mL of ultrapure water, the solution was placed in a 1 mm quartz CD cuvette, and the CD spectrum was scanned in the far ultraviolet range (190–260 nm). Distilled water was used from the dissolved samples as a blank space. The scanning rate was 500 nm/min, the response time was 1 s, the spectral bandwidth was 2.0 nm, and the resolution was 0.5 nm.

The CD spectroscopy data were averaged–smoothed and preprocessed using Spectra Manager software V2.15 and uploaded to the Dichroweb website (http://dichroweb.cryst.bbk.ac.uk, accessed on 15 December 2020) and saved, with a wavelength range of 190 to 240 nm selected to calculate the relative content of secondary structures in CONTINLL.

#### 2.4.4. Protein Solubility

To determine the solubility of WMPHs and WMPHs–COS under different pH conditions, the methods described previously by Rodríguez-Ambriz and Martínez-Ayala were used [24]. WMPHs and WMPHs–COS were prepared in a 10 mg/mL sample solution. The pH of the sample solution was adjusted to 3.0, 4.0, 5.0, 6.0, 7.0, 8.0, 9.0, or 10.0. The supernatant was centrifuged at 4500 rpm at 4 °C for 20 min. A BCA protein assay kit was used to determine the protein concentration in the supernatant.
Solubility (%) = (M_0_/M) × 100(2)
where M_0_ is the protein content in the supernatant and M is the total protein content in the sample.

#### 2.4.5. Emulsifying Ability and Emulsion Stability

The emulsifying ability and emulsion stability were determined according to previous methods described by Molina, Papadopoulou [25]. An amount of 1.5 mL of the WMPHs or WMPHs–COS (10 mg/mL) was mixed with 0.5 mL of walnut oil and homogenized at room temperature with a high-speed disperser (3000 rpm, 3 min). An amount of 4.9 mL of 0.1% SDS solution was diluted with 100 μL of emulsion and mixed, after which the absorbance was measured at 500 nm. After standing for 30 min, the absorbance at 500 nm was measured to determine the emulsion stability using the same method.
Emulsifying ability = (2 × T × A_0_ × N))/(c × Φ × 10,000)(3)
Emulsion stability = A_0_ (A_0_ − A_30_) × 30(4)
where T = 2.303; N is the dilution ratio of the emulsion; c is the concentration of protein in the protein aqueous solution (g/mL); Φ = 0.25 is the volume fraction of walnut oil in the original emulsion; A_0_ and A_30_ are 0 min and 30 min, respectively; and the absorption of the emulsion is stable at 500 nm.

### 2.5. Preparation and Process Optimization of WMPHs–COS–Ca

The WMPH solution was divided into 5 components using tubes with ultrafiltration membranes with different pore sizes (<1 kDa, 1–3 kDa, 3–5 kDa, and >5 kDa). WMPHs–COS were prepared in a solution (2% *w*/*v*), and CaCl_2_ solution was added to screen for the components with the best calcium chelation rate. The influence factors of WMPHs–COS/CaCl_2_ ratio (2:1, 3:1, 4:1, 5:1, and 6:1 *w*/*w*), pH (5, 6, 7, 8, 9, and 10), temperature (30, 40, 50, 60, and 70 °C), and time (40, 50, 60, 70, and 80 min) were investigated. The calcium chelation rate was used as an index to investigate the influence of various factors on the chelation reaction.
Calcium chelation rate (%) = (W − W_0_)/W(5)
where W is the total calcium content in the sample and W_0_ is the calcium content in the supernatant.

### 2.6. Structure Characterization

#### 2.6.1. Scanning Electron Microscopy (SEM)

At room temperature, the samples were directly bonded to the conductive adhesive (dry method), the excess powder sample was purged, and a layer of gold film was plated in the vacuum spraying instrument with an average current of 15 mA and a vacuum degree of 7–8 Pa. After approximately 120 s, the sample was removed, placed under a scanning electron microscope to observe the shape and surface characteristics, and then photographed and recorded.

#### 2.6.2. UV–Vis Spectroscopy

WMPHs, WMPHs–COS, and WMPHs–COS–Ca were dissolved in deionized water (1 mg/mL). The spectra were recorded using a UV–Vis spectrophotometer (UV-6100S; Yuanxi, China) with a wavelength range of 190–800 nm [26].

#### 2.6.3. Fourier Transform Infrared (FTIR) Spectroscopy

WMPHs, WMPHs–COS, and WMPHs–COS–Ca were mixed with 20 mg of dry KBr, ground, and loaded on an FTIR instrument (NicoletiS10; Thermo, USA). The spectrum of each sample was scanned from 4000 cm^−1^ to 400 cm^−1^ [26].

### 2.7. Stability under Different Temperatures, pH Values, and Simulated Digestion Conditions In Vitro

#### 2.7.1. Temperature Stability

The methods used were described previously by Wu W [27]. WMPHs–Ca or WMPHs–COS–Ca (1 g) were dissolved in 200 mL of deionized water and mixed, and the chelating solution was divided into 5 parts. The water bath was adjusted to 50 °C, 60 °C, 70 °C, 80 °C, 90 °C and 100 °C for 60 min, after which the mixture was precipitated with a 5-fold volume of ethanol. The content of free calcium in the supernatant was determined. The temperature stability was determined using the calcium retention rate, which was calculated as follows:Calcium retention rate (%) = (x_1_ − x_2_)/x_1_ × 100(6)
where x_1_ is the total calcium content and x_2_ is the calcium content in the supernatant.

#### 2.7.2. pH Stability

The methods used were described previously by Wu W [27]. First, 1.0 g of WMPHs–Ca or WMPHs–COS–Ca was dissolved in 200 mL of deionized water, and the chelate solution was mixed and divided into 5 parts. The pH of the chelate solution was adjusted to 2.0, 4.0, 6.0, 8.0, or 10.0, and the mixture was allowed to react at 37 °C for 60 min. A 5-fold volume of ethanol precipitation was used to determine the content of free calcium in the supernatant. The calcium retention rate was calculated using the formula given in Section 2.7.1.

#### 2.7.3. Simulated Digestion In Vitro

To evaluate the stability of the WMPHs, WMPHs–COS–Ca, and WMPHs–Ca during the digestion process, in vitro gastrointestinal digestion was simulated according to the INFOGEST 2.0 method [28]. The methods used were described previously by Xiao J [29] and slightly modified.

In the stomach digestion stage, 1 g of WMPHs–Ca or WMPHs–COS–Ca was weighed; 8 mL of gastric digestive reserve fluid, 0.42 mL of pepsin (2000 U/mL), and 27.5 μL of CaCl_2_ solution (44.1 g/L) were added; the pH was adjusted to 3.00; and the volume of ultrapure water was adjusted to 10 mL. After mixing, the samples were placed on a constant temperature shaking table at 37 °C for 2 h, after which the samples were removed after gastric digestion.

During the intestinal digestion stage, 4 mL of intestinal digestive reserve liquid, 2.5 mL of pancreatic enzyme (100 U/mL), 1.5 mL of pig bile salt (200 mg/mL), and 20 μL of CaCl_2_ solution were added; the pH was adjusted to 7.00; the volume of ultrapure water was set to 20 mL; the mixture was thoroughly mixed; and the mixture was placed on a constant temperature shaking table at 37 °C to avoid light for 2 h. The mixture was centrifuged at 4 °C (10,000 rpm, 10 min), and the supernatant was collected and stored at −20 °C until further analysis [18]. The calcium retention rate was calculated using the formula in Section 2.7.1.

### 2.8. Statistical Analysis

Origin 2021, GraphPad 9.5, and Design-Expert 8.0.6.1 were used for graphic processing. SPSS 23 was used for the data analysis. Statistical significance was determined via Duncan’s multiple range test of one-way ANOVA. *p* < 0.05 was considered to indicate statistical significance. All measurements were performed thrice in parallel, and all data are presented as the mean ± standard deviation.

## 3. Results

### 3.1. Preparation and Process Optimization of WMPHs–COS

#### 3.1.1. Optimization of the Extraction Process Parameters in the One-Factor Experiment

The effect of the WMPH/COS ratio is shown in Figure 1A. The WMPH/COS ratio was 1:3, and the glycation rate reached a maximum (64.02%). Due to the high saccharide content, the degree of browning in the WMPHs–COS was the highest, with an absorbance of 0.806. After the Maillard reaction, ketone and aldehyde intermediates and small-molecule active products can polymerize or bind peptides to form melanin and brown material, which is unfavorable for the production of Maillard reaction intermediates, leading to a decrease in the reaction rate [30]. When the WMPH/COS ratio was 1:1, the degree of browning was low, and the glycation degree was 53.08%. Overall, a fixed WMPH/COS ratio of 1:1 was used to explore the effect of concentration on the preparation of WMPHs–COS, as shown in Figure 1B. As the concentration of WMPHs increased from 15 to 25 mg/mL, the glycation and browning degree increased. The highest glycation degree was observed at 25 mg/mL (61.38%). Additionally, with increasing peptide concentration, the color of the WMPHs–COS solution became darker than that at low concentrations, and the degree of browning gradually increased. Therefore, a concentration of 15 mg/mL WMPHs is recommended for preparing optimal glycation products and reducing costs.

pH, temperature, and time also affect the glycation reaction. The effect of pH on WMPHs–COS preparation is shown in Figure 1C. As the pH increases, the glycation and browning degrees decrease. At a pH of 8.0, the glycation degree was 54.94%, and the browning degree was 0.305. At a pH of 10.0, the lowest glycation degree was 46.83%, which was attributed to the degradation of the protein under alkaline conditions [31]. Temperature is one of the critical factors affecting the Maillard reaction (Figure 1D). In the range of 50–60 °C, the glycation reaction progresses rapidly. With increasing temperature, the protein chain extends, exposing the original amino group, which is embedded in the molecule, to the molecular surface, and the Maillard reaction occurs slowly [32]. Therefore, in the 60–90 °C range, the Maillard reaction rate was slow, and considering the degree of browning, 80 °C was chosen for subsequent experiments. The effect of time on the preparation of WMPHs–COS is shown in Figure 1E. With increasing time, the glycation and browning degrees gradually increased, and the Maillard reaction was relatively complete. After 6 h, 55.01% of the proteins were glycated, and the degree of browning significantly differed from that at the other time points (*p* < 0.05).

#### 3.1.2. Optimization of Glycation Conditions by RSM

According to the results of the one-factor experiment, 29 optimization experiments were performed with the glycation degree (*Y*) as the response variable, including time (*A*), concentration of WMPHs (*B*), pH (*C*), and temperature (*D*) as the four factors (Table 1 and Table 2). Furthermore, the data were fitted to a quadratic polynomial regression equation, and the polynomial equation of the glycation degree (Y) was as follows:Y = 62.68 + 0.18*A* + 4.46*B* − 0.96*C* + 2.56*D* + 5.59*AB*
− 3.59*AC* − 7.98*AD* + 2.22*BC* − 1.64*BD* − 1.77*CD* − 4.50*A*^2^
− 4.27*B*^2^ − 4.31*C*^2^ − 10.11*D*^2^(7)

We used ANOVA to assess the validity of the quadratic polynomial model. The ANOVA results are shown in Table 3. According to a regression test of *p* < 0.01 and a regression model test of *p* = 0.3779 > 0.05, the proposed regression equation has less error when fitting the experimental data, and the independent variables have a significant impact on the results [33]. The coefficient of determination *R*^2^ = 0.9216, the adjustment coefficient *R*^2^adj = 0.8433, the Adeq Precision was 12.210, and the coefficient of variation (*CV*) was 5.83%, indicating that the test was reliable and accurate [34], and the study results satisfied the model reproducibility requirements.

Figure 2 shows the 3D surface response plots. When the contour is elliptical or saddle-shaped, the interactive effect between two factors is significant [35]. With increasing slope on the *AB* and *AD* surfaces, the interactive effect was highly significant (*p* < 0.01), the *AC* interactive effect was significant (*p* < 0.05), and the *BC*, *BD*, and *CD* interactive effect were not significant (*p* > 0.05). According to the *p* value, the order of influence of each factor on the glycation degree was as follows: concentration of WMPHs (*B*) > temperature (*D*) > pH (*C*) > time (*A*). The effect of the interaction on the glycation degree decreased in the order *AD > AB > AC > BC > CD* > *BD*. Through RSM optimization, the optimal glycation conditions were determined to be a duration of 7.80 h, a WMPH concentration of 19.08%, a pH of 7.29, a temperature of 76.5 °C, and a predicted degree of glycation of 65.75%.

To ensure the reliability and authenticity of the optimization results, we adjusted the relevant parameters based on the operation of the reagents. The optimal process parameters for WMPH–COSs were determined to be as follows: a time of 8 h, a WMPH concentration of 20%, a pH of 7.3, and a temperature of 77 °C. With these parameters, the glycation degree was 64.82%, which was 0.93% different from the predicted value, indicating the model’s reliability and practical application value.

### 3.2. Structure and Processing Characteristics of the WMPHs and WMPH–COSs

#### 3.2.1. SDS–PAGE

SDS–PAGE can be used to determine the distribution of the molecular weights of glycation products on a map [23]. After SDS–PAGE, Coomassie blue staining was performed to determine the molecular weight of the WMPHs. As shown in Figure 3A, the molecular weight of the WMPHs was approximately 10 kDa. However, after reacting with COS, the molecular weight of the WMPHs increased, and the WMPHs were distributed above 180 kDa.

#### 3.2.2. Circular Dichroism (CD) Spectroscopy

The CD spectrum was used to determine the secondary structure of the WMPHs–COS. As shown in Figure 3B, the WMPHs–COS had a distinct peak between 190 and 205 nm, with the lowest molar ellipticity occurring at 198 nm, indicating structural modification of the WMPHs with COSs. The hydrogen bonds between watermelon peptide hydrolysates (WMPHs) mainly maintain the secondary structure. Hydroxyl (–OH) or amine (–NH2) groups in COSs react with residues on the peptide chain, –CO–NH–, or amino acids, to form new chemical bonds. This reaction destroys the original hydrogen bonds, leading to a change in the secondary structure of WMPHs [36]. The content of the secondary structure in the WMPHs and WMPHs–COS samples was calculated by Dichroweb (http://dichroweb.cryst.bbk.ac.uk, (accessed on 15 December 2020)). WMPHs and WMPHs–COS primarily consisted of β–sheets and random coils, with these structures constituting over 70% of their composition. After glycation, the number of α–helix and random coil structures decreases, and the number of β–sheets increases, transforming WMPHs to a disordered structure, which is due to the exposure of the internal group after the protein is connected to the saccharide chain, resulting in a spatial conformation change [37].

#### 3.2.3. Amino Acid Composition

Peptide glycation results in the formation of a copolymer of peptides and saccharides through covalent bonding between the amino acid in the peptides and the reduced carbonyl group in the reducing saccharides. The Maillard reaction reduces the number of amino and carbonyl groups present over time [38]. The relative amino acid contents of the WMPHs and WMPHs–COS are shown in Table 4. The relative contents of Lys and Arg in the WMPHs–COS decreased by 0.46% and 1.06%, respectively, compared with those in the WMPHs. These two amino acid side chains contain free amino groups, which can covalently bind to the hydroxyl group of saccharides [39]. Similarly, compared with those in WMPHs, the relative contents of Asp, Ser, and Tyr in WMPHs–COS decreased by 2.07%, 1.35%, and 3.15%, respectively. Thiol and hydroxyl groups are in the side chains of the above amino acids and can react with saccharide molecules [40].

#### 3.2.4. Protein Solubility of the WMPHs and WMPHs–COS

The degree of solubility directly affects emulsification, gelation, foaming, and other processing properties, and the pH strongly influences the solubility of WMPHs [41]. The solubilities at different pH levels are shown in Figure 3C. When the pH was 5.0, the solubility was lowest when the WMPHs flocculated. However, compared with those of WMPHs, the solubility of WMPHs–COS increased by 14.52% at the isoelectric point, and hydrophilic groups were produced, leading to improved solubility [42]. Because the amino acids in the WMPHs reacted with the carbonyl group in saccharides and more hydrophilic groups were introduced, WMPHs–COS were more soluble than WMPHs in the pH range of 3.0–10.0, indicating that WMPHs–COS have better solubility and potential for application in product processing.

#### 3.2.5. Emulsifying Ability and Emulsion Stability of WMPHs and WMPHs–COS

In product processing, better emulsification can significantly reduce the interface tension between oil and water and promote stable emulsion formation [43], as shown in Figure 3D. Due to the introduction of hydrophilic groups on saccharide chains and changes in peptide structure, some exposed hydrophobic residues increase the adsorption capacity of oil molecules during the Maillard reaction. The amino acids in solution are quickly transferred to the interface between the oil and water, promoting emulsion formation [44]. Compared with that of the WMPHs, the emulsification of the WMPHs–COS was 10.16 mg^2^/g greater (*p* < 0.0001). In addition, the emulsification stability of the WMPHs–COS improved from 204.33 min to 257.06 min (*p* < 0.001), which was due to the increased steric hindrance between the peptides and saccharides. Moreover, the peptides’ molecular junction fragment and the side chain in the graft had a certain hydrophobicity, allowing them to be quickly and tightly and evenly absorbed at the oil/water interface, improving the emulsification stability of the WMPHs [45].

### 3.3. Preparation of WMPHs–COS–Ca

WMPHs were classified as fractions of different molecular weights (<1 kDa, 1–3 kDa, 3–5 kDa, and >5 kDa) to further investigate the effect of molecular weight on the calcium chelation rate (Figure 4A). The calcium chelation rate of WMPHs with a molecular weight <1 kDa was the highest (64.88%) and was significantly different from that of peptides >5 kDa (*p* > 0.05), indicating that WMPHs (<1 kDa) were more likely to chelate calcium. WMPHs–COS were prepared from WMPHs (<1 kDa) according to Section 2.3.1.

Different process conditions can affect the chelating ability of calcium. In this study, we selected the WMPHs–COS/CaCl_2_ ratio, temperature, time, and pH to explore the optimal preparation process for WMPHs–COS–Ca (<1 kDa). In addition to the experimental variables, the process conditions were as follows: a WMPHs–COS/CaCl_2_ ratio of 5:1, a temperature of 60 °C, a time of 60 min, and a pH of 7.0. As shown in Figure 4B, the calcium chelation rate increased slowly after the WMPHs–COS/CaCl_2_ ratio reached 5:1 and reached the highest rate when the WMPHs–COS/CaCl_2_ ratio was 6:1 (65.53%). However, the addition of excessive peptide caused the saturation of calcium ion chelation, slowing the chelation rate.

Temperature and pH are important factors affecting the calcium chelation rate. When the temperature increased from 30 °C to 70 °C, the calcium chelation rate tended to increase first and then decrease (Figure 4C), showing that the chelation reaction of calcium ions and peptides is an endothermic process and that increasing the temperature facilitates the generation of chelates [46]. In the temperature range of 30 °C–50 °C, the binding of WMPHs–COS to calcium ions increased, with the highest chelation rate being 63.19%. The calcium chelation rate decreased when the temperature was greater than 60 °C, indicating that a high temperature is not conducive to the chelation reaction. Due to changes in the peptide structure, the molecule can bind calcium ions but also curls inward, which reduces its capacity to bind calcium ions [47]. pH significantly affects the chelation rate (Figure 4D), increasing the coordination ability of Ca^2+^ with –NH_2_ and –COOH in the pH range of 5.0–8.0 [48]. The most favorable chelation rate was observed at pH 7.0 (64.10%). With increasing pH, the chelation rate decreased due to the ease with which calcium ions generate Ca(OH)_2_ precipitates under alkaline conditions, reducing the amount of calcium bound to the polymer [49]. For the chelation time (Figure 4E), the chelation rate was the highest (63.76%), suggesting that WMPHs–COS reacted rapidly with calcium. However, with increasing chelation time, the Ca^2+^ concentration gradually reaches saturation [27], and the reaction rate decreases.

Based on the above results, WMPHs <1 kDa were selected for use in preparing WMPHs–COS, and the optimized preparation conditions were as follows: a WMPHs–COS/CaCl_2_ ratio of 5:1, a temperature of 50 °C, a chelation time of 50 min, and a pH of 7.0.

### 3.4. Structural Characterization

#### 3.4.1. UV–Vis Spectroscopy

The energy required for the electronic transition of groups changes the maximum absorption wavelength and absorption intensity of UV–vis spectroscopy [50]. As shown in Figure 5A, the specific groups, carboxyl groups, and amide bonds of the WMPHs had absorption peaks at approximately 291 nm. After the Maillard reaction, the maximum absorption peak of the WMPHs–COS redshifts to 316 nm, which causes changes in the conformation of the protein due to glycation and the exposure of hydrophobic groups, resulting in increased UV absorption of aromatic amino acids [51]. The formation of metal ions and organic ligands can cause new or original absorption peaks to move in UV–vis spectroscopy. The WMPHs–COS–Ca peak appears at 300 nm. A redshift occurs due to the conjugation effect between π electrons on –NH_2_ and –OH and carbon-based double bonds, resulting in a reduced transition energy of n → σ* [52]. These results suggest that WMPHs–COS can bind to calcium ions to form WMPHs–COS–Ca.

#### 3.4.2. Fourier Transform Infrared (FTIR) Spectroscopy

The characteristic changes in FTIR absorption peaks are commonly used to analyze chemical structures, especially the interaction of metal ions with organic ligand groups [53]. As shown in Figure 5B, after binding with COSs, the FTIR absorption peak shifted, and the intensity of the peak changed.

Compared with that of WMPHs–COS, the wavenumber of WMPHs–COS–Ca shifted from 3378.82 cm^−1^ to 3362.42 cm^−1^, which was attributed to the –NH stretching vibration corresponding to the Ca^2+^ binding site, suggesting that N–H plays a role in chelate formation, possibly due to the dipole field effect or induction effect [54]. The vibrational spectral region at 1700–1500 cm^−1^ corresponds to the stretching vibration of the amide I band (1700–1600 cm^−1^; C=O) and the amide II band (1580–1510 cm^−1^; C–N,N–H) [55]. The peak at 1662.89 cm^−1^ shifted to 1658.07 cm^−1^, indicating a change in the C=O absorption peak and suggesting that C=O is involved in the formation of WMPHs–COS–Ca, which aligns with the findings of Wu, He [27]. The N–H band of the amide II band shifted from 1580.93 cm^−1^ to 1562.61 cm^−1^, suggesting the involvement of N–H in Ca^2+^ chelation. Additionally, the shift of 1402.05 cm^−1^ to 1412.66 cm^−1^ indicates the stretching vibration of the carboxylic acid group (–COO–) and suggests that –COO– may undergo a chelation reaction to convert to –COO–Ca [56]. The peak at 1153.00 cm^−1^ was shifted to 1260.78 cm^−1^, corresponding to the amide III band. This band represents the stretching vibration of C–N and the bending vibration of N–H, which mainly arise from the combination of –CH_2_ in glycine with proline and calcium ions [55]. The peak at 625.33 cm^−1^ was shifted to 614.72 cm^−1^, indicating the substitution of the N–H bond by N–Ca.

#### 3.4.3. Scanning Electron Microscopy (SEM)

SEM is commonly employed for analyzing the microstructure of materials. We examined the WMPHs, WMPHs–COS, and WMPHs–COS–Ca samples via SEM. The results are shown in Figure 5C. The surface of the WMPHs appears to have a smooth and thin structure. Compared with the WMPHs surface, the WMPHs–COS surface became thicker and had smaller irregular fragments attached to it. After chelating with calcium, the surface of the WMPHs–COS–Ca became loose with irregular depressions, transitioning from flakes to numerous granular aggregates, which was attributed to the coordination between the WMPHs–COS and calcium ions. The carboxyl and amino groups combine with Ca^2+^ to create a “bridging effect”, resulting in alterations in the internal structure and the formation of a compact chelate structure [57].

### 3.5. Stability under Different Temperatures, pH Values, and Simulated Digestion Conditions In Vitro

#### 3.5.1. Temperature Stability

The impact of temperature on protein processing is significant. Figure 6A shows the calcium retention rates of WMPHs–Ca and WMPHs–COS–Ca at various temperatures. The calcium retention rate of WMPHs–COS–Ca was comparable to that of WMPHs–Ca at 50 °C (*p* > 0.05). This result aligns with the observations of Wu, He [27]. The calcium retention rate of WMPHs–Ca was greater than that of WMPHs–COS–Ca at 60 °C (*p* < 0.0001), which can be attributed to heat treatment at 80 °C during the glycation process, which disrupted the spatial structure of the peptide and released calcium ions. Consequently, as the temperature increased, the binding ability of WMPHs–COS–Ca decreased, but the calcium retention rate still exceeded 90%. These results highlight the impressive resistance of the peptide–calcium chelates to heat.

#### 3.5.2. pH Stability

Figure 6B shows the stability of peptide–calcium chelates at different pH values. WMPHs–COS–Ca remained stable in the pH range of 2.0–10.0. However, the retention rate of WMPHs–Ca significantly decreased in the pH range of 2.0–4.0 (*p* < 0.05) because the high concentration of H^+^ under acidic solid conditions can compete with calcium ions for the active binding group, causing the peptide–calcium chelates to dissociate [58]. There was no significant difference between the WMPHs–COS–Ca in the pH 2.0 range and those in the pH 4.0 range (*p* > 0.05), suggesting that glycation improved the calcium ion dissociation of the peptide–calcium chelates under strongly acidic conditions. In the pH range of 6.0–10.0, as the solution environment became alkaline, the concentration of OH– increased, resulting in reduced competition between OH– and metal ions for active binding groups. The retention of calcium in WMPHs–Ca and WMPHs–COS–Ca was greater under alkaline conditions than under acidic conditions (*p* < 0.05), suggesting that glycation enhances the stability of peptide–calcium chelates in alkaline environments. An increase in the rate of calcium chelation in the intestine prevents precipitation and facilitates efficient absorption by intestinal epithelial cells [59]. These results indicated that WMPHs–COS–Ca may enhance calcium sequestration in the gastrointestinal tract.

#### 3.5.3. Simulated Digestion In Vitro

Disadvantageous factors (H^+^ and protease) can induce calcium release in the gastrointestinal tract, where insoluble precipitates and Ca(OH)_2_ are formed, resulting in low calcium bioavailability. The stability of the calcium chelate under a simulated gastrointestinal pH is illustrated in Figure 6C. Most calcium is released in the simulated stomach environment, which is not favorable for intestinal absorption. However, compared to those in the control group and in the WMPHs–Ca group, the calcium retention rate in the WMPHs–COS–Ca group was greater in the simulated gastric environment.

In simulated intestinal fluid, an alkaline environment inhibits the release of calcium ions from WMPHs–COS–Ca and promotes chelation between peptides and Ca^2+^ [27]. Consequently, the calcium retention rate in the simulated intestinal fluid was greater than that in the control group, indicating that WMPHs–COS–Ca exhibit better tolerance to pepsin and pancreatin.

## 4. Discussion

In this study, glycated peptides and a glycated peptide–calcium chelate were prepared, and the processing characteristics of the resulting glycated walnut meal hydrolysate and the structure and digestion characteristics of the resulting glycated peptide–calcium chelate were determined. These results indicate that WMPHs–COS–Ca could be a promising new calcium supplement.

In the Maillard reaction, the saccharide/peptide ratio, peptide concentration, pH, temperature, and time influence WMPHs–COS formation [14]. A high content of saccharides leads to an increased chance of interaction between molecules, which may cause a caramelization reaction [60]. In addition, in the single-factor experiment and response surface optimization analysis of variance, temperature was an important factor affecting the formation of the glycated peptides (*p* < 0.05). With increasing temperature, molecular movement significantly accelerates, exposing more binding sites and saccharide molecular combinations, which is conducive to the Maillard reaction [61] (Figure 1D).

The structure and processing properties (solubility and emulsification) of the glycated peptides were characterized by SDS–PAGE, circular dichroism spectroscopy and amino acid analysis. As shown in Figure 3A, after glycation, the bands of the sample moved upward and the color decreased, which was due to the reaction of the sugar carbonyl group in the COSs, resulting in an increase in molecular weight. The decrease in band color was mainly due to protein aggregation or cross-linking induced by the Maillard reaction [62]. As shown in Figure 3B, the percentages of α-helix and random coil structures decreased by 1.2% and 6.6%, respectively, and the percentage of β-sheet structures increased by 3.9% after glycation. Due to the tighter and more stable nature of the β-sheets and the random coils compared to the α-helixes, the structure of WMPHs–COS is more flexible than that of WMPHs, suggesting that WMPHs–COS have better processing properties [63]. In addition, amino acid analysis revealed that the content of the glycated peptide aspartic acid was lower than that of the glycated peptides. Notably, Asp has a free carboxyl group that can bind to calcium ions [64], indicating the potential of WMPHs–COS to form glycated peptide–calcium chelates.

According to the FTIR results, the WMPHs exhibited a broad absorption peak at 3340.31 cm^−1^, which became narrower and shifted to 3378.82 cm^−1^ after glycation, indicating an –OH stretching vibration. Similarly, the peak at 2960.42 cm^−1^ shifted to 2921.72 cm^−1^, indicating the occurrence of a C–H bond stretching vibration upon glycation. The –OH and C–H bonds are characteristic peaks of saccharides [65], demonstrating that the Maillard reaction of WMPHs occurs with COSs. Additionally, the peak at 1640.73 cm^−1^ shifted to 1662.89 cm^−1^, which is associated with changes in protein secondary structure [66], suggesting an increase in β-sheets and a decrease in random coils. These findings are consistent with the CD spectrum results in this study (Figure 3B). After chelating with calcium ions, the tensile vibrations of the amide I band, the amide II band, and the amide III band corresponding to the bonding bond were detected (C=O, N–H, and –COO). Based on the above results, the WMPHs–COS and calcium binding sites included amino nitrogen atoms, carboxylic oxygen atoms, and carbon-based oxygen atoms.

Finally, gastrointestinal simulation results showed that glycation also slows the release of peptide–calcium chelates in simulated gastric fluid, allowing more calcium to enter the intestine and thereby improving calcium bioavailability. After simulated digestion in the gastrointestinal tract, trace amounts of calcium ions were released from WMPHs–Ca and WMPHs–COS–Ca, suggesting that calcium ions can be absorbed as soluble chelators, increasing calcium absorption, similar to the findings of a previous study [27]. Similar results were also observed for chelates prepared from casein phosphopeptides and calcium, which showed that phosphopeptides could maintain a good ability to chelate calcium under alkaline conditions so that calcium could exist in the intestine in a soluble form and that the time required for calcium to be digested by the human body was prolonged [67].

## 5. Conclusions

In this study, glycation was utilized to prepare WMPHs–COS and WMPHs–COS–Ca. The findings revealed the formation of macromolecular glycoproteins in WMPHs–COS, resulting in a change in the secondary protein structure. Glycation improved the solubility and emulsification activity of the resulting products, significantly enhancing the processing performance of the WMPHs. Chelation of WMPHs–COS with calcium resulted in a chelation rate of 65.43%, and the results showed that chelation caused structural transformation of WMPHs–COS and that the presence of amino nitrogen atoms, carboxyl oxygen atoms, and carbon oxygen atoms was involved in glycation and chelation reactions. In addition, WMPHs–COS–Ca exhibited excellent stability with respect to temperature, pH, and simulated gastrointestinal liquid, demonstrating the potential to regulate calcium release in the gastrointestinal environment and improve bioavailability. Future research should investigate the calcium absorption-promoting effect of the glycated walnut peptide–calcium chelates from a biological perspective and study their calcium absorption and transport mechanism. This study can be used as an important reference for new functional calcium supplements.

## Figures and Tables

**Figure 1 foods-13-01109-f001:**
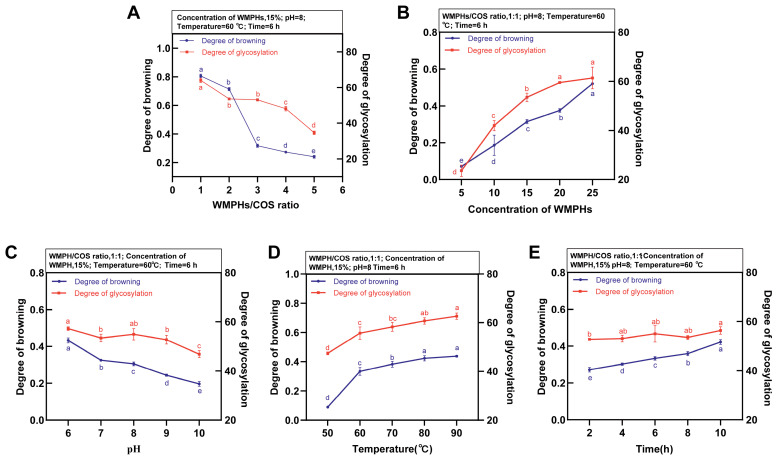
Effects of the WMPH/COS ratio (**A**), concentration of WMPHs (**B**), pH (**C**), temperature (**D**), and time (**E**) on the degree of glycation. Different letters indicate significant differences.

**Figure 2 foods-13-01109-f002:**
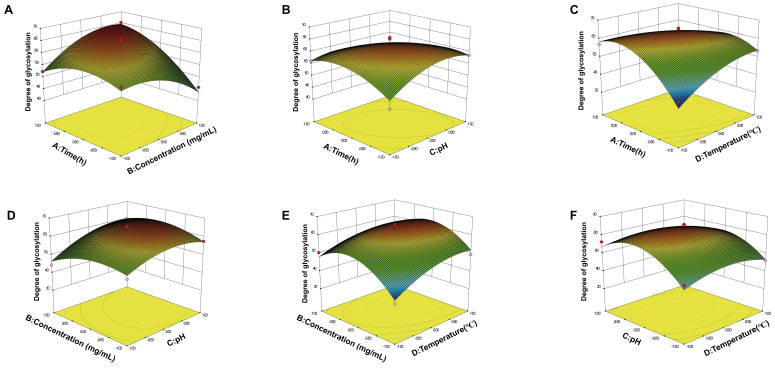
Response surface plots of the effects of four factors on the glycation degree. (**A**) Interaction between *A* (time) and *B* (concentration of WMPHs). (**B**) Interaction between *A* (time) and *C* (pH). (**C**) Interaction between *A* (time) and *D* (temperature). (**D**) Interaction between *B* (concentration of WMPHs) and *C* (pH). (**E**) Interaction between *B* (concentration of WMPHs) and *D* (temperature). (**F**) Interaction between *C* (pH) and *D* (temperature).

**Figure 3 foods-13-01109-f003:**
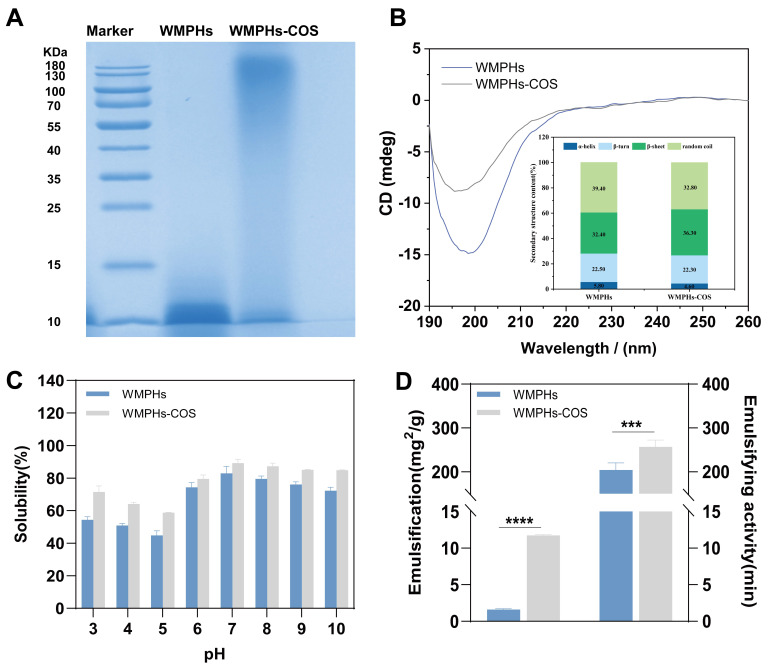
Structural and processing characteristics of the WMPHs and WMPHs–COS. (**A**) SDS–PAGE electrophoresis. (**B**) CD spectrum. (**C**) Protein solubility. (**D**) Emulsifying ability and emulsion stability. The symbols indicate statistical significance: ***: *p* ≤ 0.001, ****: *p* ≤ 0.0001.

**Figure 4 foods-13-01109-f004:**
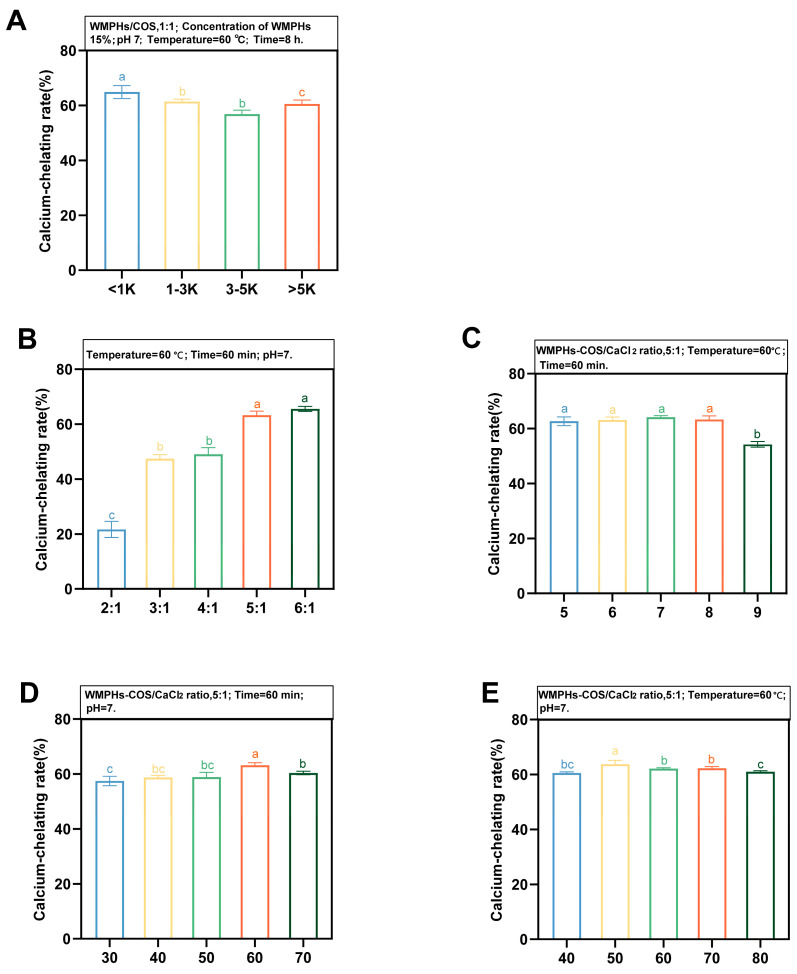
Effects of the molecular weight of the WMPHs (**A**), WMPH/CaCl_2_ concentration (*w*/*w*) (**B**), pH (**C**), temperature (**D**), and time (**E**) on the calcium-binding capacity. Different letters indicate significant differences.

**Figure 5 foods-13-01109-f005:**
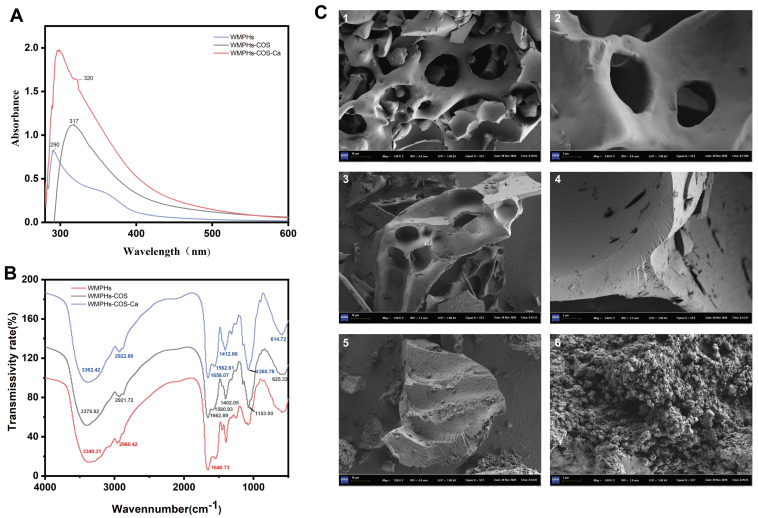
Structural characterization of WMPHs, WMPHs–COS, and WMPHs–COS–Ca. (**A**) UV–vis absorption spectrum. (**B**) FTIR spectrum. (**C**) Scanning electron microscopy. 1–2: WMPHs, 3–4: WMPHs–COS; 5–6: WMPHs–COS–Ca.

**Figure 6 foods-13-01109-f006:**
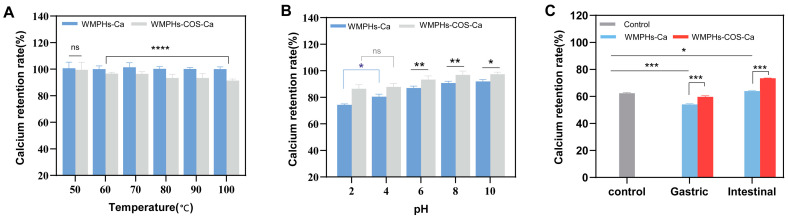
Stability of WMPHs–Ca and WMPHs–COS–Ca. (**A**) Temperature stability. (**B**) pH stability. (**C**) Simulated gastrointestinal digestion. The symbols indicate statistical significance: ns: *p* > 0.05, *: *p* < 0.05, **: *p* < 0.01, ***: *p* < 0.001, ****: *p* < 0.0001.

**Table 1 foods-13-01109-t001:** Response surface optimization level.

Level	Factors			
	*A* Time (h)	*B* Concentration of WMPHs (mg/mL)	*C* pH	*D* Temperature (°C)
−1	4	10	7	70
0	6	15	8	80
1	8	20	9	90

**Table 2 foods-13-01109-t002:** Response surface test scheme and results.

Number	*A* Time (h)	*B* Concentration (mg/mL)	*C* pH	*D* Temperature (°C)	*Y* Glycation Degree (%)
1	6	10	8	70	37.75 ± 0.54
2	6	10	7	80	50.97 ± 2.47
3	4	15	8	70	40.64 ± 0.59
4	8	15	8	70	53.31 ± 0.26
5	6	15	8	80	65.75 ± 1.34
6	6	20	8	70	49.30 ± 0.44
7	6	15	9	90	48.06 ± 1.16
8	6	20	8	90	55.37 ± 0.74
9	4	15	9	80	56.19 ± 0.96
10	4	15	8	90	56.84 ± 0.12
11	8	15	8	90	37.61 ± 1.54
12	6	20	7	80	57.48 ± 0.17
13	6	10	8	90	50.38 ± 0.59
14	4	15	7	80	47.85 ± 1.76
15	6	15	8	80	62.27 ± 0.13
16	6	15	8	80	65.12 ± 0.43
17	8	15	7	80	58.53 ± 0.44
18	4	10	8	80	56.68 ± 1.30
19	4	20	8	80	52.33 ± 0.34
20	6	15	7	90	56.29 ± 0.42
21	6	15	8	80	60.35 ± 2.10
22	8	15	9	80	52.50 ± 0.38
23	6	20	9	80	59.71 ± 1.01
24	6	10	9	80	44.34 ± 0.36
25	6	15	7	70	47.00 ± 1.78
26	6	15	8	80	59.92 ± 0.15
27	8	10	8	80	45.67 ± 1.83
28	6	15	9	70	45.86 ± 0.74
29	8	20	8	80	65.14 ± 2.03

**Table 3 foods-13-01109-t003:** Analysis of variance.

Source	Sum of Squares	df	Mean Square	F Value	*p* Value Prob > F	
Model	1574.87	14	112.49	11.76	<0.0001	Significant
*A*–Time	0.41	1	0.41	0.043	0.8392	
*B*–Concentration of WMPHs	238.89	1	238.89	24.98	0.0002	
*C*–pH	10.97	1	10.97	1.15	0.3023	
*D*–Temperature	78.46	1	78.46	8.2	0.0125	
*AB*	141.84	1	141.84	14.83	0.0018	
*AC*	51.59	1	51.59	5.39	0.0358	
*AD*	254.47	1	254.47	26.61	0.0001	
*BC*	19.64	1	19.64	2.05	0.1738	
*BD*	10.74	1	10.74	1.12	0.3072	
*CD*	12.55	1	12.55	1.31	0.2712	
*A* ^2^	131.49	1	131.49	13.75	0.0023	
*B* ^2^	118.45	1	118.45	12.38	0.0034	
*C* ^2^	120.76	1	120.76	12.63	0.0032	
*D* ^2^	663.14	1	663.14	69.33	<0.0001	
Residual	133.9	14	9.56			
Lack of Fit	105.31	10	10.53	1.47	0.3779	not significant
Pure Error	28.59	4	7.15			
Cor Total	1708.77	28				

**Table 4 foods-13-01109-t004:** Relative amino acid compositions in the WMPHs and WMPHs–COS.

Amino Acid	Relative Contents	
	WMPHs	WMPHs–COS
Asp	8.11 ± 0.16	6.04 ± 0.16
Thr	2.52 ± 0.32	1.82 ± 0.19
Ser	4.53 ± 0.15	3.17 ± 0.06
Glu	17.82 ± 0.93	13.47 ± 0.98
Gly	3.63 ± 0.65	4.00 ± 0.12
Ala	3.01 ± 0.01	2.90 ± 0.13
Cys	0.47 ± 0.01	0.65 ± 0.08
Val	5.59 ± 0.53	5.91 ± 0.28
Met	1.86 ± 0.06	4.31 ± 0.41
Ile	3.27 ± 0.32	3.79 ± 0.57
Leu	6.78 ± 0.26	4.83 ± 0.81
Tyr	3.34 ± 0.46	0.18 ± 0.01
Phe	4.31 ± 0.59	24.61 ± 0.59
His	2.05 ± 0.05	1.75 ± 0.28
Lys	2.03 ± 0.01	1.57 ± 0.27
Arg	2.95 ± 0.07	1.89 ± 0.44
Pro	8.19 ± 0.24	6.07 ± 0.33

## Data Availability

The original contributions presented in the study are included in the article, further inquiries can be directed to the corresponding authors.

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
