# Peer review of "Glycated Walnut Meal Peptide–Calcium Chelates: Preparation, Characterization, and Stability"

_foods, 2024, doi:10.3390/foods13071109_

Round 1

Reviewer 1 Report

Comments and Suggestions for Authors

Abstract - It seems to this reviewer that there are some outstanding results that deserve to be very briefly cited here. Please check.

Introduction - It contains a good level of information.

Objectives - They are clear.

Techniques - They are described properly, however there are several techniques (i.e., 2.7.1, 2.7.2) that lack references. Authors are invited to correct this lack of information.

Conclusions - They are rather poor. Authors need to indicate briefly, but very clearly, where to go from here; where to go in future research works based on the results obtained in this interesting investigation. These comments are very important according to this referee.

References - There are not less than two recent,, and relatively recent, scientific reports, related to this work which deserve to be cited in order to improve the information to potential readers. Authors are invited to take into account this suggestion.

Finally, I hope authors take into consideration the cited suggestions. This report is a good and very useful piece of work. Congratulations to authors! 

Figure captions - There are abbreviations 

Reviewer 2 Report

Comments and Suggestions for Authors

Foods-2911239

Title: Glycosylated walnut meal peptide-calcium chelate: Preparation, characterization and stability

The paper’s idea is innovative and interesting. I have, however, some suggestions described below. I also suggest that authors improve the discussion section with more comparisons with other studies and discuss further the impact of their observed changes.

Specific comments

1.     Please write all keywords different from the ones in the title.

2.     Line 39: Please include in which nutrients is it rich on.

3.     In this first paragraph, please add numbers (production, percentage of oil and meal, etc.)

4.     Line 46: Why do the authors attribute proteins from walnuts as “high-quality”? Please add more information about amino acid composition to support this declaration.

5.     Line 63: Please add also the word “glycation” to represent the Maillard reaction. In some biology books, glycation is the non-enzymatic reaction (such as Maillard) and glycosylation refers to the enzymatic reaction between proteins and sugars.

6.     Line 82: Please add the type of Coomassie blue dye used.

7.     Line 84: Please add more information about the chitosan oligosaccharides (size distribution, how they were extracted and from each source are they). This information is necessary for other authors to repeat the glycation experiment and to understand the structure of the prepared chelate.

8.     Line 90: HCl, not HCL.

9.     Line 95: Which alkaline protease at which enzymatic activity? This information is crucial to repeat enzymatic assays because activity of enzymes change from batch to batch.

10.   Line 102: How much COS and WMPHs were added to the system in mass? How come this is not disclosed?

11.   Lines 127-129: Why is this information here?

12.   Line 243: “T samples”?

13.   Line 155: There is always a space between the number and the unit.

14.   Line 157: This assay proposes to access the protein solubility. I suggest the title is changed to “protein solubility”.

15.   Line 166: “Emulsibility” is not recognized as a word in many dictionaries. I suggest the authors keep to “Emulsifying ability and emulsion stability”.

16.   Line 179: The action of “dividing a specific solution 4 components” is denominated “Fractionation”. What do you mean by “ultrafiltration tube”? Ultrafiltration cartridges? Please specify the brand used as well as centrifuging conditions.

17.   Line 234: Have the authors checked the distribution of data and if it complied with ANOVA’s assumptions? Please specify the names of the tests to check for normality, homoscedasticity, etc.

18.   Figure 1: Images are too small. Please try to add just two figures per line. I also suggest that the authors paint the colour of the respective results the titles of the axis. In this way they can remove the legends.

19.   Line 240: The information on the ratio of WMPHs and COS should be given in the Material and Methods section. Also please precise if it refers to a mass ratio.

20.   Lines 257-272: It is not that clear why the authors have studied the effect of pH, temperature and time in the glycosylation level as these are well studied factors for the Maillard reaction. Similar effects are expected and are already reported elsewhere. Please explain.

21.   Section 3.2.1. There are many critics in the literature about adding time as an independent factor when the result (in this case glycosylation) depends on a time-dependent reaction. What happens is that the rate of the reaction is necessarily dependent on the said “independent” factor. It also alters the weight that the other factors receive because it “bends” the mathematical model toward the overweighted time factor. This can be clearly seen in the results: interactions between other factors apart from time are non-significant; moreover, the A2 interaction is very high. If time was not included, that would most probably not be the case. I suggest the authors are more careful in the next experiments and do not add time as a factor. One possible strategy is to do the statistical optimization and then a kinetics study.

22.   Figure 3 is unfortunately not clear enough (resolution). Please improve it. I can barely see the content of Figure B.

23.   Line 508: In the beginning of the paper the authors refer to the reaction as “glycosylation”. I already made a suggestion to prefer the term “glycation”. However, here in the discussion the authors use the term glycation. Please unify “glycation” throughout the paper, including title.

24.   Line 514: Please verify if “collision” is the best word choice here.

25.   Line 522: “circular binary chromatography”???

26.   Lines 514-530: Please work on the writing of this section. The readers do not know results from which analysis you’re discussing when the authors say for example “the bands moved upward”.

27.   Lines 531-533: This finding is common in studies that compare glycosylated peptides with non-glycosylated ones. Please verify.

Comments on the Quality of English Language

They are in the specific comments to authors.

Reviewer 3 Report

Comments and Suggestions for Authors

Dear Authors,

Please find attached some suggestions for your manuscript.

This draft need some improvements.

There are many misinterpretations and speculations.

Kind regards

Round 2

Reviewer 3 Report

Comments and Suggestions for Authors

Indeed, they improve the quality of manuscripts but I still have remarks for authors

a. line 363 carbonyl group (not carbanyl)

b. Figure 5 Panel A. Again the absorbance for WMPHs-COS-Ca is almost 2. This graph should be corrected! It is not allowed to have an absorbance higher than 1-1.2!!!
